# Predictive Value of MUC5AC Signature in Pancreatic Ductal Adenocarcinoma: A Hypothesis Based on Preclinical Evidence

**DOI:** 10.3390/ijms24098087

**Published:** 2023-04-30

**Authors:** Ashish Manne, Anup Kasi, Ashwini Kumar Esnakula, Ravi Kumar Paluri

**Affiliations:** 1Department of Internal Medicine, Division of Medical Oncology at the Arthur G. James Cancer Hospital and Richard J. Solove Research Institute, The Ohio State University Comprehensive Cancer Center, 460 W 10th Ave, Columbus, OH 43210, USA; 2Medical Oncology, The University of Kansas Medical Center, 2330 Shawnee Mission Pkwy, Westwood, KS 66025, USA; 3Department of Pathology, The Ohio State University Wexner Medical Center, 460 W 10th Ave, Columbus, OH 43210, USA; 4Section of Hematology and Oncology, Department of Medicine, Wake Forest School of Medicine, 475 Vine St, Winston-Salem, NC 27157, USA

**Keywords:** pancreatic ductal adenocarcinoma, MUC5AC, biomarker, predictive, gemcitabine, chemoresistance

## Abstract

Mucin 5AC (MUC5AC) glycoprotein plays a crucial role in carcinogenesis and drug sensitivity in pancreatic ductal adenocarcinoma (PDAC), both individually and in combination with other mucins. Its function and localization are glycoform-specific. The immature isoform (detected by the CLH2 monoclonal antibody, or mab) is usually in the perinuclear (cytoplasmic) region, while the mature (45 M1, 2-11, Nd2) variants are in apical and extracellular regions. There is preclinical evidence suggesting that mature MUC5AC has prognostic and predictive (response to treatment) value. However, these findings were not validated in clinical studies. We propose a MUC5AC signature with three components of MUC5AC—localization, variant composition, and intensity—suggesting a reliable marker in combination of variants than with individual MUC5AC variants alone. We also postulate a theory to explain the occurrence of different MUC5AC variants in abnormal pancreatic lesions (benign, precancerous, and cancerous). We also analyzed the effect of mature MUC5AC on sensitivity to drugs often used in PDAC management, such as gemcitabine, 5-fluorouracil, oxaliplatin, irinotecan, cisplatin, and paclitaxel. We found preliminary evidence of its predictive value, but there is a need for large-scale studies to validate them.

## 1. Introduction

Pancreatic ductal adenocarcinoma (PDAC) is one of the aggressive cancers without significant progress on the therapeutic front for a long time [1]. Recently presented NAPOLI-3 results showed the survival advantage of nano-liposomal irinotecan-based therapy (NALIRIFOX) over gemcitabine/nab-paclitaxel (Gem-NP) [2]. It could be the standard of care in selected patients, but it may not change overall PDAC outcomes, as the median overall survival is similar to that from the PRODIGE trial (11.1 months) with FOLFIRINOX. Unfortunately, immune checkpoint inhibitors and targeted therapies have not yet revolutionized PDAC management, as they have in other cancer types, such as breast and lung cancer. 

One of the inherent challenges in the management of PDAC is the lack of effective predictive biomarkers to help guide the selection of chemotherapy regimens, such as 5FU-based regimens, e.g., FOLFIRINOX or NALIRIFOX, versus gemcitabine-based regimens, such as Gem-NP. *BRCA* mutations can help identify patients who may benefit from platinum-based treatments, but these mutations are rare in PDAC (<5%) [3]. Currently, the decision of treatment is based on functional status and comorbidities that may not reflect the potential response to a particular chemotherapeutic regimen. Having an effective biomarker in this regard is helpful to gear treatment in a personalized approach. In this review, we focus on a novel biomarker, MUC5AC, that has the potential to improve PDAC management. 

Mucin 5AC (MUC5AC) is a gel-forming, glycosylated, high-molecular-weight protein expressed in abnormal pancreatic tissues, including PDAC [4,5]. In our previous publication, we discussed different MUC5AC variants and reviewed evidence on the prognostic value of immature MUC5AC (detected by anti-mucin 5AC CLH2 monoclonal antibody, or mab), which was inconclusive [6]. Post-transcriptional changes of MUC5AC in pancreatic cells, specifically N-glycosylation, promote carcinogenesis, as demonstrated in a study by Pan et al. [7]. Such changes include following a sequence of steps, starting from the perinuclear region, to the apical region, dimerization of the unglycosylated MUC5AC monomer, the addition of N-acetyl galactosamine residues (maturation by glycosylation), multimerization, and finally secretion of mature MUC5AC into the duct or inter-cellular regions [8].

MUC5AC isoforms can be broadly divided into immature and mature MUC5AC variants for practical purposes [9,10,11,12,13,14,15,16,17]. Immature MUC5AC is the initial un- or less glycosylated variant in the perinuclear/cytoplasmic region. It can be detected by the CLH2 mab. Mature MUC5AC is a heavily glycosylated MUC5AC variant detected by 45M1 or 2-11M1 or Nd2 mabs, and is localized primarily in apical, extracellular (secreted or inter-cellular) regions. When subjected to growth factors, pancreatic cancer cell lines (PCLs) produced more mature than immature MUC5AC isoforms, indicating the difference in their function and malignant potential [17]. Immunogenic MUC5AC refers to a subtype of mature MUC5AC variant with an epitope capable of eliciting an immune reaction detected by NPC-1C and PAM4 mabs [9,10]. Some studies also identified other immunogenic peptides, such as MUC5AC-A02-1398 (FLNDAGACV) and MUC5AC-A24-716 (TCQPTCRSL) on MUC5AC that can provoke cell-mediated toxicity [18]. The CLH2 mab targets the sequence TTSTTSAP within the tandem repeat (backbone) of MUC5AC. It can recognize glycosylated and unglycosylated MUC5AC variants, except when galactosamine residues cover this region. The epitopes of 45M1 (C-terminal), 21-M1 (N-terminal), and Nd2 are on glycosylated MUC5AC, but can partially react to immature forms [19,20,21]. Therefore, it is safe to conclude that glycosylation, localization, immunoreactivity to specific mabs, and the function of MUC5AC are interlinked. 

In this paper, we introduce the concept of a ’MUC5AC signature for PDAC’, discuss MUC5AC’s role in carcinogenesis and promoting distant metastases, and discuss its regulation. We examined the available indirect evidence of MUC5AC’s predictive value in the PCLs. 

## 2. MUC5AC Signature

Inaguma et al. proposed that MUC5AC ‘sorting’ (distribution to apical vs. extracellular vs. perinuclear or cytoplasmic) is well controlled in gastric/respiratory cells, where it is normally produced [22]. The control of such a process is lost in lung cancers, cholangiocarcinoma, and PDAC. This does not explain what triggers MUC5AC expression and why it does not transform all pancreatic cells into cancerous ones (MUC5AC is detected in some benign diseases, such as intraductal papillary mucinous neoplasms or IPMN). Spinning off from this hypothesis, we propose the following sequence of events that can happen in the malignant transformation of pancreatic cells and the acceleration of malignant disease, ultimately shaping the MUC5AC signature. 

The series of events can be broadly divided into three interlinked stages—trigger response, malignant transformation, and malignant acceleration—with three components in each stage—sorting, MUC5AC variant composition, and level or intensity of expressed MUC5AC. We did not consider MU5AC mRNA in this theory, as the maturation, or acquired immunogenicity, is a post-translational change, and mRNA expression level may not make a significant difference. However, sorting and MUC5AC variants (through glycosylation) may be related. MUC5AC is believed to support/promote malignant transformation and acceleration of metastasis [22]. Figure 1 illustrates the proposed theory based on our hypothesis. We acknowledge that multiple factors, including mutations, initiate and promote carcinogenesis. In this model, we referred to them as triggers. 

The proposed sequence of events can explain the detection of MUC5AC in benign, precancerous, and cancerous lesions; a differential pattern of expression and variants of MUC5AC in cancerous vs. precancerous forms; the malignant transformation of the latter to the former; and may explain MUC5AC’s prognostic and predictive value. Based on this hypothesis, we propose the concept of classifying PDAC by the MUC5AC signature, which incorporates three key elements of MUC5AC: composition of MUC5AC variant identified; localization of the MUC5AC variant (sorting); and intensity of the expression (Figure 2). 

### Strengthening Our Hypothesis Using MUC5AC Signature

To prove our model with available evidence in the literature, we start with two interlinked components: sorting and MUC5AC variants. There are limited studies that compare immunohistochemical staining of mature and immature MUC5AC among the spectrum of pancreatic tissues, ranging from normal, to pancreatitis, PanIN, IPMN, and tumors. CLH2 mab was used in most of the studies. Based on localization, we drew conclusions on the MUC5AC isoform. To date, there are no studies that examined the staining of NPC1C and PAM4 in benign/precancerous pancreatic pathologies. 

Kim et al. published a study in 2002 that supports sorting and the MUC5AC variant components discussed here [16]. Staining was predominantly apical and extracellular with mature (21 M1/Nd2 mab) and perinuclear/cytoplasmic with immature (CLH2 mab) MUC5AC across PanINs and cancers, proving that they exist in precancerous (PanIN and IPMN) and PDAC tissues. In another study, immature MUC5AC (by CLH2 mab) expression pattern and frequency were compared in PanIN lesions derived from surgical specimens from patients with adjacent normal (N-PanIN) and PDAC (C-PanIN) tissues. CLH2 staining was cytoplasmic (perinuclear), as well as apical/extracellular in both N-PanIN and C-PanIN, indicating possible cross-reactivity of CLH2 mab with mature MUC5AC [23]. The frequency of immature MUC5AC was significantly different in PanIN-1A (*p* < 0.0001) and PanIN 2 (*p* < 0.005) lesions in N-PanIN and C-PanIN patients. Interestingly, there were no morphological differences among the PanIN groups, suggesting a minimal effect of MUC5AC production on the morphology (discussed in Section 4.3). The staining pattern for immunogenic MUC5AC is similar to that of mature MUC5AC (and different from immature) [9,10,11,12]. Finally, MUC5AC produced when stimulated by growth factors such as vasoactive peptide (VIP), or transcription factors (TFs) such as GLI1, is primarily apical or extracellular, further supporting the sorting process (discussed in Section 3 below) [17,22].

The intensity of staining was compared between PanIN and PDAC in a study by Ochinuda et al. [24]. Strong MUC5AC expression was seen in 48% of PDACs, vs. 0% of PanINs. Moderate expression was reported in 35% of PDACs vs. 65% of PanINs, and weak in 17% of PDACs vs. 36% of PanINs. None of the samples were negative. Overall, strong/moderate expression was 83% in PDAC vs. 64% in PanINs. Compared to non-tumoral tissues, malignant tissues had a 2.8-fold greater expression of mRNA, but this differential expression was not statistically significant. Immature MUC5AC staining was focal and cytoplasmic in PanIN, but it was strongly positive in PDAC in another study [25]. PanIN and atypical duct areas in non-neoplastic tissues (with negative staining) showed strong MUC5AC expression, hinting at the initiation of pre-malignant changes with MUC5AC expression. 

The MUC5AC signature for the PDAC model can explain the wide range of outcomes reported by studies that used CLH2 mab, which we summarized in our previous publication [6]. Most studies reported cytoplasmic staining, and patients were classified positive or negative based on the extent of staining, making it difficult to assess the prognostic value of the MUC5AC signature. When the thresholds to classify were low (5 or 10%) immature MUC5AC expression was a good prognostic marker [26,27]. Alternatively, when the thresholds were high (25% or high-H score), the outcomes were poor with its detection [28,29]. This underscores the importance of using all three components (isoform, localization, and intensity) in the proposed MUC5AC signature. 

## 3. Regulation of MUC5AC Expression

In PDAC, the influence of MUC5AC can be simplified, as shown in Appendix A. It stimulates tumor cell proliferation and distant metastasis, protects the tumor from host defenses, and reduces sensitivity to chemotherapeutic agents, such as gemcitabine (gem) [22,30,31,32,33]. The regulation of MUC5AC production is not well understood, but two important regulators identified are the growth factors (GFs) and TFs (Figure 3). 

MUC5AC gene expression is constitutively repressed in normal pancreatic cells and, therefore, the respective protein is not detected in them [34]. Epigenetic silencing, by CpG methylation and H3 Lysine 9 modification of the promoter, is believed to be one of the principal processes of such suppression [35,36]. Supporting this theory, Yamada et al. reported higher methylation levels in MUC5AC-deficient compared to MUC5AC-positive lines [36]. Interestingly, demethylation of the CpG sites immediately proximal (5′) to the MUC5AC gene transcription site by 5-aza-2’-deoxycytidine (5AZA) could not increase MUC5AC production in a preclinical study published in 2003 [37]. This was confirmed by other studies, and could mean that MUC5AC gene expression is controlled by other sites/mechanisms, and epigenetics is not the primary mechanism [35]. 

Kato et al. later identified two such regions corresponding to binding sites for TFs, specificity protein 1 (Sp-1) and activator protein 1 (AP-1), as regulatory sites for MUC5AC gene transcription [38]. Both of the TFs participate in basal MUC5AC production in a malignant pancreatic cell, while AP-1 also takes part in Phorbol 12-myrisate 13-acetate (PMA)-induced MUC5AC production. PMA stimulation phosphorylates sub-units of AP-1 (cFos/cJun) via protein kinase C (PKC)/ERK/AP-1 and PKC/JNK/AP-1 pathways that, in turn, upregulate MUC5AC promoter activity (to produce mRNA). ERK and JNK inhibitors downregulate this process, suggesting their role in MUC5AC regulation. 

Likewise, Krüppel-like zinc-finger protein GLI-1 also promotes MUC5AC production by activating its gene promoter [22]. GLI-1-induced MUC5AC’s function seems to be different from that of Sp-1- or AP-1-induced MUC5AC in the following ways: (1) it reduces the amount of E-cadherin at the intercellular membranes, thus promoting cell migration; and (2) it increases the nuclear accumulation of beta-catenin and excess target gene expression, thus promoting cellular invasion. GLI-1 expression correlates with MUC5AC expression in PanIN and PDAC tissues (low to undetectable in PanIn-1A/1B and high in PanIn-2/3 and PDAC). GLI-1′s expression statistically correlated with altered E-catherin (loss of membrane localization) and beta-catenin (increased nuclear localization) in PanIN-3 and PDAC tissues. GLI-1 does not appear to be responsible for malignant transformation, but some critical aberrations frequently associated with PDAC carcinogenesis, such as KRAS mutations and irregularities in the hedgehog signaling pathway, promote its production as well [39,40,41]. 

GFs are known to promote pancreatic cell growth, migration, and invasion. GFs such as forskolin and VIP also promote MUC5AC production and release in malignant pancreatic tissues [17,42,43,44,45]. They exert their effect through cyclic adenosine monophosphate (cAMP)-dependent kinases, also known as protein kinase A (PKA), just as epidermal growth factor receptors (EGF) are overexpressed in PDAC [46,47]. In vitro studies have even demonstrated enhanced VIP-induced MUC5AC production in the presence of EGF [43]. A study published in 2006 illustrated that GF-induced MUC5AC is mature, with the following observations: (1) In an unstimulated PCL, the CLH2 mab stained immature variants in the perinuclear region intensely, while the mature (Nd2) mab stained both perinuclear and cytoplasmic (peripheral) regions. When stimulated by forskolin, CLH2 continued to stain the perinuclear region, but mature mabs (45 M1, Nd2, and 21 M1) stained the mucins that accumulate in apical and extracellular regions (even between cells). (2) The VIP-induced release of mature MUC5AC is more on the cells’ luminal (apical) side than on the basolateral side. Interestingly, basolateral VIP receptors appear to significantly affect VIP-induced MUC5AC when stimulated or inhibited (by PKA inhibitors). (3) In PDAC tumor tissue, CLH2 could stain only in the cytoplasm, while 45M1, Nd2, and 2-11M could stain both luminal and cytoplasmic regions. CLH2 noticeably did not stain the mucin in the lumen (secreted). Therefore, it can be deduced that GF-induced MUC5AC is predominantly heavily glycosylated, or promotes glycosylation. Interestingly, after forskolin-induced MUC5AC production, inhibition of O-glycosylation increased the immunoreactivity of the CHL2 mab in PCLs, but there was no change for other mabs, further strengthening the concept that N-glycosylation is important in carcinogenesis [7]. 

In summary, MUC5AC is not expressed in a normal pancreatic cell, likely due to epigenetic silencing. The expression is mediated via PKA/PKC pathways, with GF and TF as key regulators; however, the etiology of the triggers that start and propagate its production in pancreatic cells is unknown. Detection of GLI-1 and GNAS mutations that alter mucin gene expression profiles in pancreatic ductal cells, including the overexpression of MUC5AC through the MAPK/PIK3 pathway in precancerous lesions such as pancreatic intraepithelial neoplasia (PanIN) and IPMNs, may be an early indicator of malignant transformation [22,48]. If we extrapolate the evidence from other organs, such as the lung, nasal epithelium, and gastrointestinal tract, they can be infectious (bacterial or viral), inflammatory (through cytokines), environmental (smoking), or growth factors (epidermal growth factor) [49,50,51,52,53,54,55,56,57]. This warrants close examination of the pancreatic tumor microenvironment that plays a role in MUC5AC production or upregulation. 

## 4. Predictive Value of MUC5AC

To understand the impact of MUC5AC on treatment response, we used the results of the preclinical studies reported by Krishn et al. as a reference [58]. This study proved MUC5AC’s role in the pancreatic cancer cell, including viability, anchorage-independent growth, motility, adhesion to the extracellular matrix, angiogenesis, apoptosis, and sensitivity to gemcitabine [58]. Through in vivo studies in nude mice, the authors suggested its role in promoting tumor growth, metastasis, and disease progression (Table 1). For immunohistochemistry (IHC) and western blot (WB), mature MUC5AC, 45M1 mab was used. 

Important takeaways from this study are as follows: (i) Mature MUC5AC is detected in most PDACs, and not detected in normal pancreatic tissues. (ii) All PCLs did not express mature MUC5AC. (iii) There is discordance between MUC5AC mRNA and protein expression in some PCLs. (iv) Inhibiting MUC5AC could improve outcomes, including sensitivity to gemcitabine. To further verify the observations of this study, we compared cell growth, migration, invasion, and drug sensitivity of PCLs with wide-ranging levels of MUC5AC. The goal of this analysis was to assess the influence of mature MUC5AC on PDAC. 

### 4.1. Comparing PCLs Based on MUC5AC Expression

PCLs were classified based on their native MUC5AC protein and/or mRNA expression (Table 2). Among the studied PCLs, there were PCLs with concordance between mRNA and protein expression, as in COLO 357 (CO), SW 1990 (SW), BxPc3 (B), and MIA PaCa (MPC); discordant PCLs were the medium mRNA expressors with low or no protein expression, as in PANC-1 (P) and AsPc3 (A). Alternatively, in T3M4 PCL, there is high mRNA expression, but low protein expression (protein expression of SU 8686 was not reported). This matches observations in clinical samples with PanINs and PDAC, and CLH2-reactive MUC5AC; the discordance was high in PanIN1b (45% for in-situ hybridization vs. 82% IHC), PanIN2 (81% vs. 94%), and PDAC (61% vs. 100%) [16]. The concordance was 100% in PanIN3 lesions, while the discordance was minimal in PanIN1A (62% vs. 59%). To assess the influence of mature MUC5AC on various aspects of PDAC, aspects of malignant cells, including cell growth, migration, invasion, and drug sensitivity, were compared based on their expression levels (Table 3).

### 4.2. Comparing Basic Pathological Characteristics

MUC5AC protein expression and basic characteristics are illustrated in Appendix A. There is no significant correlation between MUC5AC expression levels and basic characteristics of PCLs, such as source (primary tumor vs. metastatic), doubling time, and differentiation. PDACs stained by mature (21M1 and Nd2 mab) and immature (CHL2 mab) mabs did not differ morphologically in a retrospective study published two decades ago, supporting this observation [16]. In that study, 21M1 mab stained 100% of well-differentiated tumors, while CLH2 and Nd-2 stained 90% and 85%, respectively. The frequency of 21M1- and CLH2-positive tumors in moderately (96% vs. 92%) and poorly (59% vs. 64%) differentiated tumors was marginal. Remarkably, there were differences in staining frequencies between two mabs that identify mature MUC5AC (21M1 and Nd2), further highlighting the influence of the other two components in the MUC5AC signature (location and intensity) in PDAC. 

### 4.3. Comparing Malignant Potential

To understand the impact of MUC5AC on the malignant potential of PCLs, COLO357 (CO) and MIA PaCa (M), representing high and no MUC5AC (mRNA and protein) expression, respectively, were selected (Table 3). The literature search was limited, as there were no head-to-head studies. We divided the search into two parts: (1) poor pathophysiological characteristics or prognostic factors, and (2) chemosensitivity or predictive factors. The results are summarized in Table 3. 

We expanded the search comparing the PCLs based on Table 2 (Table 4) [59,65,66,67,68,69,70,71,72,73,74,75,76].

### 4.4. Comparing Chemosensitivity

Widely used treatment regimens are 5FU (FOLFIRINOX, NALIRIFOX, FOLFOX, NALIRI) or gem-based (Gem-NP or GemOx). We attempted to establish the best treatment group based on mature MUC5AC expression levels. The chemosensitivities of PCLs to some of the standard drugs used in treating PDAC, such as gemcitabine (gem), fluorouracil (5FU), cisplatin (cis), irinotecan (iri), and oxaliplatin (Ox) were compared using three studies, and one PCL for each combination (mRNA and protein) (as in Appendix A) [63,64,75]. There were no studies that looked at sensitivity of NP or NALIRI. The difference in paclitaxel was too close to see any pattern in Michalski et al.’s study, and we did not include it in our analysis (Figure 3 and Table 5). 

We studied the PCLs that were similar in both studies. CO was not studied by Fujita et al. and, therefore, SW was used to represent high MUC5AC protein expressors. Gem sensitivity was too close to observe a trend from Michalski et al.’s study [63]. We relied on Fujita et al.’s study (Figure 4) for our observations.

For practical purposes, we can divide PCLs into three groups—no, high, and low–moderate expressors—for both protein and mRNA. Our analysis did not identify any correlation between MUC5AC expression and 5FU or paclitaxel sensitivity. NP and NALIR were not tested. PCLs expressing higher MUC5AC protein (western blotting) were less sensitive to gem and ox, and more sensitive to irinotecan and cis, than those which do not produce MUC5AC. Low—moderate expressors were more and less sensitive to gem and ox, respectively. For cis, their sensitivity was greater than non-expressors but lower than high-expressors (Figure 3). As mature MUC5AC is expressed in most of the PDACs, positive vs. negative classification alone is not ideal. There might be a threshold for native MUC5AC, which may identify high-risk groups or a signature of MUC5AC variants (immature and mature) that can help us predict tumor response. The mRNA expression was not useful for most drugs (except irinotecan and cis). We need larger studies on clinical samples to investigate the impact of MUC5AC expression on drug sensitivity. 

We acknowledge the limitations of this analysis; however, this suggests a good signal for future clinical studies. Quantifying/detecting protein expression in PCLs by WB is not similar to IHC on tissues [77]. We cannot identify the localization of the protein (apical vs. cytoplasmic vs. extracellular) in WB. This can explain the inconclusive results for chemosensitivity discussed above. Analyzing the impact of mature MUC5AC expression based on the localization (apical or extracellular) could provide definitive results. The thresholds for classifying PCLs into high, medium, and low expressors for protein and mRNA were not clearly defined.

## 5. Conclusions

The mucin MUC5AC is unique in the sense that its detection in pancreatic tissues is abnormal and exists in many isoforms, which can be broadly classified into mature and immature subtypes. Observations from preclinical studies need validation from retrospective or prospective studies, but provide us with a preliminary indication of the role of the MUC5AC protein as a potential biomarker in PDAC. We extensively discussed the clinical significance of serum and tissue MUC5AC in our previous publication [6]. Serum MUC5AC could be a good diagnostic marker when combined with CA 19-9, but its prognostic and predictive value is not well established [78]. Immunogenic MUC5AC, a mature MUC5AC detected by Niemann Pick C1 (NPC-1C) and PAM-4 (Clivatuzumab tetraxetan) antibodies failed to improve outcomes in PDA patients. The addition of NPC-1C antibody did not add any benefit to patients treated with gemcitabine and nab-paclitaxel in the second line after progressing on FOLFIRINOX [79]. A phase 3 trial was initiated to test the benefit of yttrium-90-labeled hPAM4 ((90) Y-hPAM4) to gemcitabine (PANCRIT-1, NCT01956812). This trial was terminated, as interim analysis did not show any clinical benefit.

The proposed MUC5AC signature for PDAC, which incorporates main isoforms (mature vs. immature), their location (linked to their function), and expression levels (the MUC5AC signature for PDAC), needs to be further validated in larger studies in order to harness its potential as an efficient predictive biomarker.

## 6. Patents

The Ohio State University is currently pursuing patent protection for the research discussed in this publication.

## Figures and Tables

**Figure 1 ijms-24-08087-f001:**
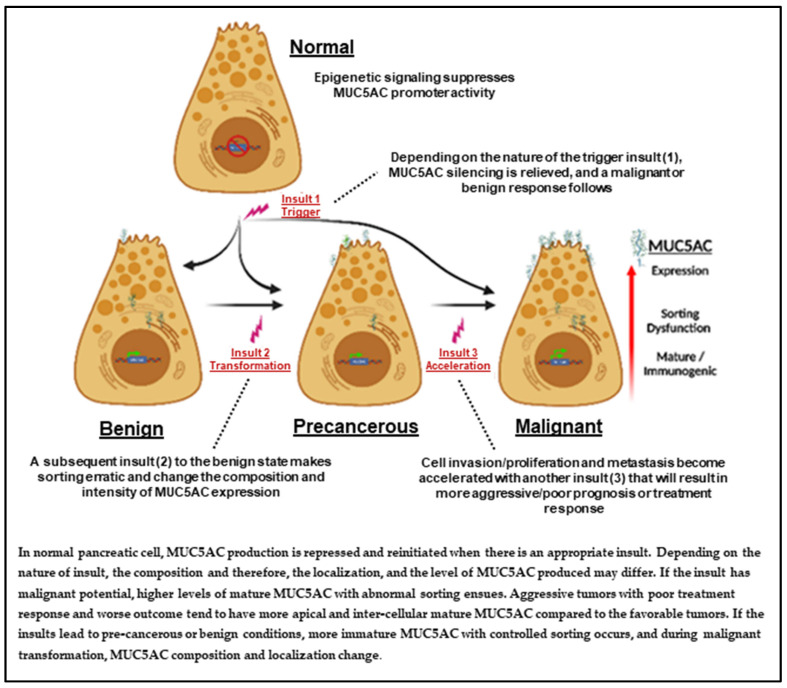
MUC5AC signature modeling in pancreatic adenocarcinoma based on the proposed hypothesis.

**Figure 2 ijms-24-08087-f002:**
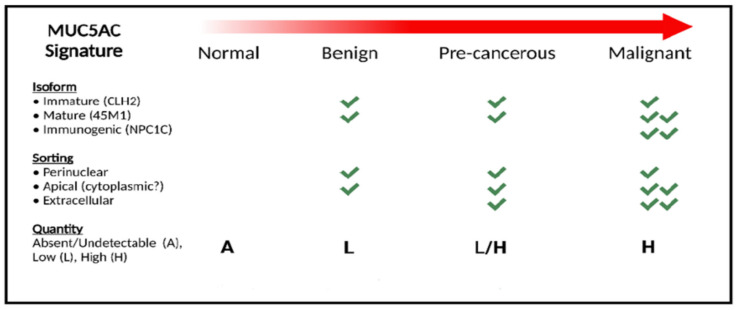
Classification of MUC5AC in pancreatic ductal adenocarcinoma.

**Figure 3 ijms-24-08087-f003:**
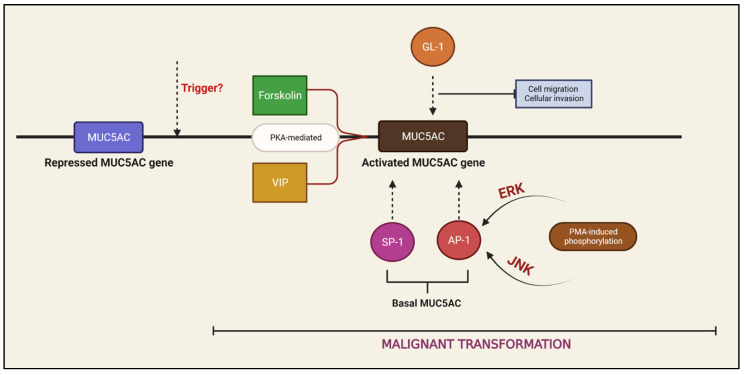
**Regulation of MUC5AC production.** VIP—vasoactive intestinal peptide; Sp-1—specificity protein 1; AP-1—activator protein 1; PMA—Phorbol 12-myrisate 13-acetate; PKC—protein kinase C; JNK—c-Jun N-terminal kinases; ERK—extracellular signal-regulated kinase.

**Figure 4 ijms-24-08087-f004:**
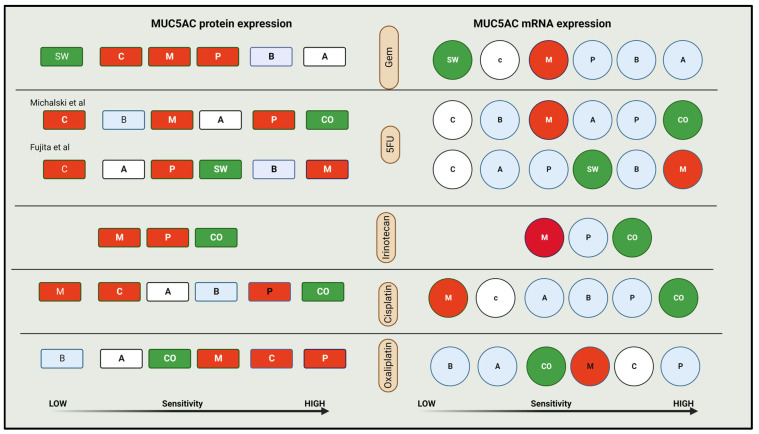
**MUC5AC expression and drug sensitivity in pancreatic cancer cell lines** [63,75]. Color coding for expression: Red—no; Green—high; Blue—Moderate; White—low; Letters—protein expression. CO—COLO357; SW—SW1990; B—BXCP3; A—ASPC-1; M—MIA PaCa; P—PANC-1; C—CAPAN-1, Gem—gemcitabine, 5FU—5 fluorouracil.

**Table 1 ijms-24-08087-t001:** Summary of MUC5AC’s role in pancreatic cell lines and in vivo studies.

Study	Results
MUC5AC expression by IHC intumor-tissues	80% of pancreatic cancer surgical specimens (positive for histology score > 0.01) Not detected in the normal tissue Localization was not reported
**In pancreatic cell lines (PCL)**	
MUC5AC-mRNA expression(by quantitative-polymerase chain reaction)	Higher in PCLs compared to normal pancreatic cell lines No expression in some PCLs (MIA PaCa, PANC10.05, QGP1)
MUC5AC protein expression(by western blot)	Positive in some PCL (ASPC1, BXPC3, COLO357, SW1990, and T3M4) Negative in some PCL (CD18/HPAF, CAPAN1, MIA PaCa, Panc-1).
Localization by confocalmicroscopy)	Stain the cytoplasm and intercellular junction (typical for mature MUC5AC)
**In MUC5AC-knockdown PCL**	
Decreased	Pancreatic cancer cell viability, anchorage-independent growth, cell motility, adhesion to the extracellular matrix, and angiogenesis Sensitivity to Gemcitabine (β-catenin mediated resistance)
Increased	Apoptosis
Nude mice with orthotopically transplanted MUC5AC knockdown PCL in pancreas *	Lesser pancreatic tumor weight Fewer metastatic sites
KrasG12D; Pdx1-Cre; Muc5ac-/- mouse models **	Delay in onset and progression of pancreatic cancer cells

* When compared, nude mice transplanted with PCL (MUC5AC positive); ** When compared to KrasG12D; Pdx1-Cre; Muc5ac+/+ mouse.

**Table 2 ijms-24-08087-t002:** Classification of select pancreatic cell lines based on MUC5AC mRNA and protein expression.

	High	Medium	Low	No
mRNA	COLO357 SW 1990 SU 8686 T3M4	BXPC3 PANC1 AsPc3	CAPAN	MIA PaCa
Protein	COLO357 SW 1990	BXPC3	AsPc3 T3M4	PANC-1 CAPAN MIA PaCa

**Table 3 ijms-24-08087-t003:** COLO357 (CO) vs. MIA PaCa (M).

	Features in CO Compared to M	Associated Features
Prognostic factors [59,60,61,62]	Higher osteopontin	Invasiveness
	Higher BMP2	Poor survival
	Higher CXCR 4 (receptor and protein)	Cell survival, proliferation, migration, invasion, and metastasis.
Predictive factors [63,64]	More sensitive to 5FU, Irinotecan, Cisplatin	
	Less sensitive to Gem and Oxaliplatin	

**Table 4 ijms-24-08087-t004:** Comparison of key characteristics in pancreatic cell lines.

Cell Lines Compared	MUC5AC Expression	Results
M vs. BxPC3	No vs. M (mRNA & protein)	BxPC3—More invasion, angiogenic potential, and tumorigenicity. More sensitive to Gem and 5FU
ASPC-1 vs. BxPC3	L vs. M (mRNA & protein)	BxPC3 has -More invasion, angiogenic potential, and tumorigenicity. More resistant to 5FU and less-similar resistance to Gemcitabine
ASPC-1 vs. CO	L vs. H (mRNA & protein)	CO—less sensitive to 5FU and more sensitive to Gem
SU86.86 vs. Panc-1	H vs. L (mRNA)	SU86.86 is less adhesive, more invasive and angiogenic potential
M vs. SW 1990	L vs. H (mRNA & protein)	SW1990 is more resistant to Gem and more sensitive to 5FU (IC 50 of 9 vs. 5.68 μM)
CO > SU86686 > BxPC3 > M	H vs. H vs. M vs. No (P) H vs. M vs. M vs. No (mRNA)	Osteopontin influenced invasiveness
BxPC-3 > AsPC-1 or M	M vs. L vs. No (P and mRNA)	Tendency to invade—MPC has a minimum tendency to invade

N—no expression; L—low expression; M—medium expression; H—high expression.

**Table 5 ijms-24-08087-t005:** Summary of drug sensitivity and MUC5AC expression.

Drug Tested	Sensitivity to the Drugs	
	Protein Expression	mRNA Expression
Gemcitabine	H < No < L-M	NDP
5FU	NDP	NDP
Irinotecan	No < H	No < L-M < H
Oxaliplatin	L-M < H < No	H < No, NDP for L-M
Paclitaxel *	NDP	NDP
Cisplatin	No < H, no NDP for L-M	No < L-M < H
Nab-paclitaxel	NT	NT
Nanoliposomal irinotecan	NT	NT

* Not discussed in the figure. H—high expression, No—no expression; L-M—low to moderate expression, NT—not tested, NDP—no definite pattern.

## Data Availability

Not applicable.

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
