# Peer review of "Predictive Value of MUC5AC Signature in Pancreatic Ductal Adenocarcinoma: A Hypothesis Based on Preclinical Evidence"

_ijms, 2023, doi:10.3390/ijms24098087_

Round 1

Reviewer 1 Report

The authors present in this paper their theory about the predictive value of MUC5AC in pancreatic ductal adenocarcinoma.

The title of the manuscript should reflect the fact that this is a hypothesis. The reader needs to know if he or she is reading a review or a hypothesis to be confirmed by further research.

The cancerization process based on MUC5AC as shown in figure 1 fully ignores the genetic mutations which are considered as the root of the transformation process. While mutations are not integrated into the hypothesis presented by the authors, this paper is fully speculative. Facts presented in fig.1 are baseless. I would suggest to fully remove this figure.

Between lines 130-150 considerations are made about the intensity of the staining with MUC5AC antibodies between different authors. The big problem is that the intensity of staining is operator-dependent and therefore, different publications can be difficult to integrate and compare.

Line 166 authors say; MUC5AC gene expression is constitutively repressed in normal pancreatic cells, and  therefore, the respective protein is not detected in them [34]. Supposedly this repression is epigenetic. However, reference 35 says: "The expression of MUC5AC was rarely influenced by epigenetic mechanisms". They need to explain this controversy.

Typo in line 42: is lact of effective predic- 42

Typo in line 47: relflect the po- 47

Typo in line 154: prognosticvalue

The usual abbreviation for Pancreatic ductal adenocarcinoma is PDAC. The authors have preferred PDA instead. It is perfectly acceptable, but it would be better for the reader if  PDAC is used.

MUC5AC: when mentioned for the first time in the manuscript, the full name should be used: MUCIN 5AC (line 52).

CLH2: when mentioned for the first time in the manuscript, the full name should be used:

Anti-Mucin 5AC antibody [CLH2] (line 55)

Lines 67 and 68 Authors say

When subjected to growth factors, pancreatic cell lines (PCLs) produced more mature than immature MUC5AC isoforms indicating the difference in their function and  malignant potential [17].

Please clarify if the pancreatic cell lines are normal cells or malignant cells. It would also be important to establish exactly which cell lines they are talking about.

Line 89

Authors say

(MUC5AC is detected in some benign diseases as well).

It would be important to mention which are those benign diseases, particularly if they can be detected in chronic pancreatitis.

Line 99: MUC5AC is believed to support/promote malignant transformation and acceleration of metastasis.

This sentence needs a reference.

tk

see comments for authors

Reviewer 2 Report

The manuscript entitled: "MUC5AC signature in pancreatic ductal adenocarcinoma: Analyzing preclinical evidence of its predictive value", by Ashish Manne is well written, structured and compiles pretty well the evidence and relevance of MUC5A in PDAC. I would like to provide some points to improve the quality of the paper.

-Edition notes:

Line 154:  “prognosticvalue” needs a space between.

-          Figure 4. MUC5AC expression and drug sensitivity in pancreatic cancer cell lines. It is difficult to clearly distinguish blue and violet properly. I would consider changing one of these colors only in figure 4. For example: violet to white.

- Regarding clinical implications of MUC5AC, a clinical trial based on mAb NPC-1C was recently published in JAMA network open, in order to determine whether NPC-1C might increase the efficacy of second-line gemcitabine and nab-paclitaxel in patients with advanced PDAC showing negative results. In my opinion, this clinical trial is crucial (since NAPOLI-3 trial which is included) and should be incorporated and discussed.

Reference:

DOI: 10.1001/jamanetworkopen.2022.49720

-As you noticed, clinical utility of MUC5AC is not only related to diagnosis. Clivatuzumab (PAM4 mAb) is being tested in a phase III clinical trial: “Phase 3 Trial of 90Y-Clivatuzumab Tetraxetan & Gemcitabine vs Placebo & Gemcitabine in Metastatic Pancreatic Cancer (PANCRIT-1)” NCT01956812.  Also, there is information available about phase Ib study with safety and partial responses reported. This information should be mentioned.

Reference:

  • DOI: 10.1016/j.ejca.2015.06.119

- There is clinical evidence about utility of MUC5AC in combination with Ca19-9 for the diagnosis of pancreatic cancer. It could help to support the preclinical evidence showed in the review and should be concisely mentioned.

References:

  • DOI: 10.1186/s12957-020-1809-z
  • DOI: 10.1038/ajg.2016.482

Is perfect

Round 2

Reviewer 1 Report

No further comments

Replace fact for lack

Author Response

Appreciate the suggestion. We changed the concerned word to 'lack'